# Computational Study on the Catalytic Performance of Single-Atom Catalysts Anchored on g-CN for Electrochemical Oxidation of Formic Acid

Abdul Qadeer [1,†], Meiqi Yang [1,†], Yuejie Liu [2,*], Qinghai Cai [1,3] and Jingxiang Zhao [1,*]

[1] Key Laboratory of Photonic and Electronic Bandgap Materials, Ministry of Education, College of Chemistry and Chemical Engineering, Harbin Normal University, Harbin 150025, China
[2] Modern Experiment Center, Harbin Normal University, Harbin 150025, China
[3] Collaborative Innovation Center of Cold Region Ecological Safety, Harbin 150025, China
* Correspondence: liuyuejie@hrbnu.edu.cn (Y.L.); zhaojingxiang@hrbnu.edu.cn (J.Z.)
† These authors contribute equally to this work.

**Abstract:** The electrochemical formic acid oxidation reaction (FAOR) has attracted great attention due to its high volumetric energy density and high theoretical efficiency for future portable electronic applications, for which the development of highly efficient and low-cost electrocatalysts is of great significance. In this work, taking single-atom catalysts (SACs) supported on graphitic carbon nitrides (g-CN) as potential catalysts, their catalytic performance for the FAOR was systemically explored by means of density functional theory computations. Our results revealed that the strong hybridization with the unpaired lone electrons of N atoms in the g-CN substrate ensured the high stability of these anchored SACs and endowed them with excellent electrical conductivity. Based on the computed free energy changes of all possible elementary steps, we predicted that a highly efficient FAOR could be achieved on Ru/g-CN with a low limiting potential of $-0.15$ V along a direct pathway of $HCOOH_{(aq)}$ $\rightarrow HCOOH^* \rightarrow HCOO^* \rightarrow CO_2^* \rightarrow CO_{2(g)}$, in which the formation of $HCOO^*$ was identified as the potential-determining step, while the rate-determining step was located at the $CO_2^*$ formation, with a moderate kinetic barrier of 0.89 eV. Remarkably, the moderate *d*-band center and polarized charge of the Ru active site caused the Ru/g-CN catalyst to exhibit an optimal binding strength with various reaction intermediates, explaining well its superior FAOR catalytic performance. Hence, the single Ru atom anchored on g-CN could be utilized as a promising SAC for the FAOR, which opens a new avenue to further develop novel catalysts for a sustainable FAOR in formic-acid-based fuel cells.

**Keywords:** single-atom catalysts; formic acid oxidation reaction; electrocatalysts; graphitic carbon nitride; density functional theory



## 1. Introduction

The direct formic acid fuel cell (DFAFC) is considered one of the most promising power sources for current and future portable electronic devices due to its high energy density, safety, portability, and secure liquid fuel storage [1–4]. For boosting the highly efficient DFAFC, it has been widely accepted that developing efficient catalysts for the formic acid oxidation reaction (FAOR) is of significance for transforming chemical energy to electrical energy [5,6]. At present, Pd and Pt materials have been extensively investigated as the most efficient and superior catalysts for the FAOR [7–12]. However, the high cost of Pt- and Pd-based catalysts due to their scarcity greatly limits their large-scale commercial applications. In addition, their low mass activity and vulnerability to CO poisoning also hamper their practical applications [13,14]. Therefore, the development of alternative FAOR electrocatalysts has been a hot topic in the research of the DFAFC in the last few decades [15–18]. Interestingly, tremendous promising strategies have been proposed for achieving the high activity and robust stability of Pt and Pd catalysts for the FAOR, such as tailoring their

electronic structures by alloying them with other elements [19–21], downsizing bulk metals into nanoparticles [22,23], and modifying their morphology and exposed facets [24–26]. Nevertheless, further innovation with a reduced amount of Pt/Pd or non-Pt/Pd is still essential to attain superior FAOR catalytic activity on low-cost catalysts.

In the last few decades, single-atom catalysts (SACs) have attracted extensive attention worldwide due to their maximized atomic efficiency, tunable and uniform active sites, and startling catalytic properties [27–32]. Interestingly, various kinds of SACs have been synthesized in recent years, such as Fe, Co, Ni, Rh, Pd, Pt, etc., which have been widely applied in electrocatalysis that are traditionally driven by expensive noble metal catalysts [33–40]. Unfortunately, however, we noted that there are few studies on the potentials of SACs for the FAOR, out of which only Li's group reported that single Rh and Ir atoms anchored on N-doped carbon exhibited high catalytic properties towards the FAOR [13,14,41], which is even superior to those of traditional Pd/C and Pt/C catalysts. Motivated by these pioneering studies, several interesting questions naturally arose: can other common SACs be utilized as FAOR electrocatalysts? If yes, which reaction mechanism is more energetically favorable for the FAOR on these SACs? How do the inherent electronic properties of these SACs affect their FAOR catalytic activity?

To address these issues, herein, by means of comprehensive density functional theory (DFT) computations, we studied the catalytic performance of a series of SACs anchored on graphitic carbon nitride (TM/g-CN) for the FAOR. It was noteworthy that some commonly synthesized SACs, including Mn, Fe, Co, Ni, Cu, Mo, Ru, Rh, Pd, Ir, and Pt, were adopted, which were extensively investigated in electrocatalysis. As for the g-CN monolayer, it can provide suitable coordination sites for SACs, resulting in strong interactions between them and the more efficient suppression of the clustering effect. In addition, the strongly polarized metal atoms bonded to N atoms can effectively boost their capability to capture FAOR intermediates. Meanwhile, N atoms receiving electrons from SACs can be activated as additional active sites to accommodate the produced H species during the FAOR dehydrogenation process. Based on the abovementioned advantages, we, thus, expected that a g-CN with uniform larger holes could be employed as an ideal substrate for SACs.

Our results showed that these TM single atoms could indeed be anchored on a g-CN monolayer with high stability and lead to excellent electrical conductivity. Furthermore, we found that the FAOR preferred to proceed along a direct pathway on TM/g-CN, in which HCOO* was identified as a key reaction intermediate with a much lower kinetic barrier of 0.04 eV than those of COOH* and CO* formation (>1.00 eV) in other pathways. More importantly, based on the computed free energy changes, we identified Ru/g-CN as a promising FAOR catalyst with a rather low limiting potential ($U_L$) of −0.15 V, which mainly came from its optimal binding strength with reaction intermediates due to its moderate d-band center and charge. Our results not only propose a new highly efficient SAC for the FAOR, but also provide a deeper understanding of the FAOR reaction mechanism, which could inspire more experimental and theoretical studies on this interesting issue.

## 2. Results and Discussion

### 2.1. Structures, Stabilities, and Properties of TM/g-CN Catalysts

Recently, g-CN was successfully fabricated via the reaction of cyanuric chloride and sodium using a simple solvothermal method [42]. Notably, different from the corrugated configuration of the well-known g-$C_3N_4$ [43–46], all atoms within the g-CN framework were in exactly the same plane (Figure S1), which is consistent with previous reports [47–52]. In other words, configurations of the $C_xN_y$ nanosheet are highly dependent on the ratio between C and N [53]. A similar planar configuration could also be observed for the $C_2N$ monolayer [54]. Furthermore, we examined the most stable configurations of these single TM atoms on the g-CN substrate by considering different possible adsorption sites (Figure 1a). After a fully structural relaxation, we found that g-CN could anchor single TM atoms within its evenly distributed N-edged cavities and maintain its planar structure. In detailed, Ni, Cu, Ru, and Pd atoms tended to be anchored by coordinating with two

adjacent N atoms, while other SACs preferred to bond with three N atoms. The shortest lengths of the formed TM–N bonds were in the range from 1.84 Å of an Fe atom to 2.22 Å of a Pt atom. Furthermore, we computed their corresponding $E_{bind}$ values as shown in Figure 1b, which exhibited considerable negative values ($-5.79$ to $-2.57$ eV), indicating the good thermodynamic stability of these TM/g-CN systems. Understandably, The strong interaction of TM atoms with g-CN could induce a significant amount of charge transfer ($0.57$–$1.27$ $|e^-|$) from a TM atom to g-CN, rendering the anchored TM atoms to carry a positive charge, which was consistent with the computed charge difference density (Figure 1c). In principle, the positively charged TM atoms could play a pivotal role as the adsorption and activation centers in catalysis.

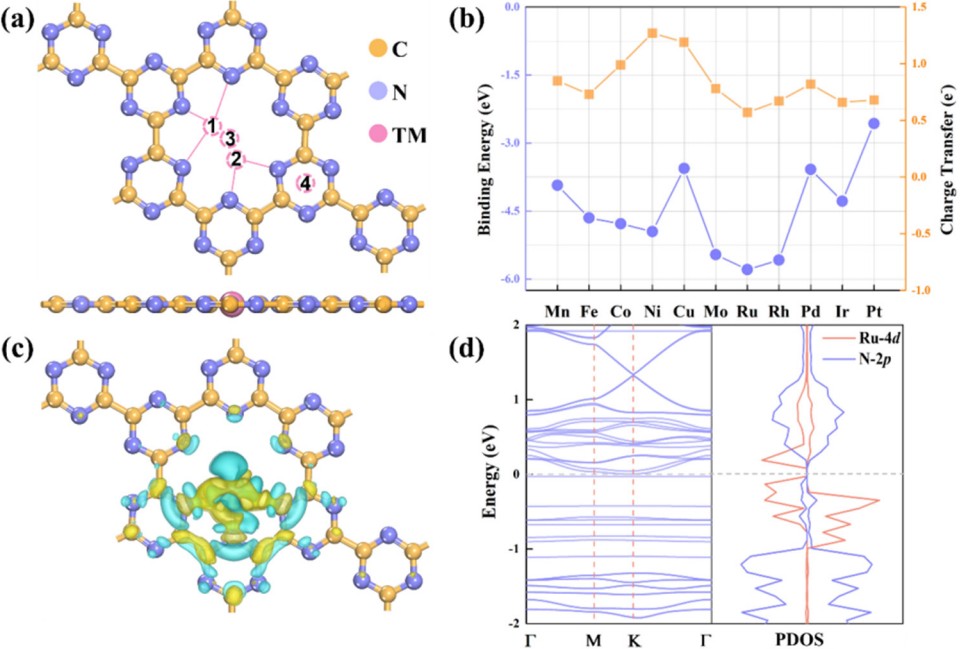

**Figure 1.** (**a**) The top and side views of TM/g-CN structure after full relaxation. (**b**) Binding energies and Bader charge transfer of different atoms on g-CN. (**c**) The charge density difference of Ru/g-CN with an isovalue of 0.003 e Å$^{-3}$. (**d**) The computed band structure and projected density of states of Ru/g-CN. The Fermi level was set to zero, indicated by the dashed line.

To further verify the stability of TM/g-CN, we performed ab initio molecular dynamics (AIMD) computations with a time step of 1.0 fs and a total time of 10 ps at T = 500 K, where Ru/g-CN was chosen as a representative due to its high FAOR activity, as previously discussed. The evolution of temperature and potential energy, as well as the final structures after 10,000 steps, are shown in Figure S2, from which we found that there was no obvious structural change for the overall frameworks and average M−N bond length throughout the entire AIMD simulation, and the metal atoms were still firmly embedded in the cavity of g-CN, suggesting their excellent dynamic stability. In addition, we examined the changes in the electronic properties by computing the energy band structures. Consistent with previous studies [47–52], the pristine g-CN is a semiconductor with a wide band gap of 3.18 eV, which would be unfavorable for transferring the charge in electrocatalytic reactions. After anchoring TM atoms, however, the band gap of g-CN greatly decreased to different degrees due to the introduction of some impurity levels (Figure S3). For example, the band gap of Ru/g-CN decreased to approximately 0.03 eV, with high peak near the Fermi level in the projected density of states (Figure 1d), indicating a significantly enhanced electrical conductivity, which could be conducive to electrochemical reactions.

## 2.2. Catalytic Performance of TM/g-CN for the FAOR

Before studying the catalytic activity of these TM/g-CN candidates towards the FAOR, we first examined the adsorption configuration of HCOOH on TM/g-CN, as the adsorption behavior of a reactant usually affects or determines the entire reaction pathway.

Since HCOOH has two isomeric structures, including trans- and cis-HCOOH, we considered their adsorption on the positively charged TM active sites of TM/g-CN. For the adsorbed trans-HCOOH*, it could be captured in an upright fashion with the carbonyl O atom binding with the TM sites, and the hydroxyl group pointing to one N atom by performing hydrogen bonding (Figure 2a). Meanwhile, the lengths of the formed TM–O bonds were in the range of 1.91~2.22 Å. In addition, we examined the HCOOH adsorption on the C and N sites and found it could not trap the HCOOH molecule, which would desorb from the C or N site or be optimized to the most stable configuration. Furthermore, we computed the adsorption energy ($E_{ads}$) of the most stable HCOOH* structures on these TM/g-CN using the formula of $E_{ads} = E_{HCOOH*} - E* - E_{HCOOH}$, where $E_{HCOOH*}$, $E*$, and $E_{HCOOH}$ represent the total DFT energies of the adsorbed HCOOH* species, pristine TM/g-CN catalyst, and isolated HCOOH molecule, respectively. Our results demonstrated that the computed $E_{ads}$ of HCOOH on these TM/–CN was in the range of −0.45 to −1.31 eV. After taking into account the correction from zero-point energy and entropy, the ΔG values for the HCOOH adsorption on these TM/g-CN monolayers turned out to be 0.27 to −0.87 eV, as shown in Figure 2b. On the other hand, the cis-HCOOH* structure generally exhibited a weaker adsorption than the trans one. For example, on Ru/g-CN, the ΔG value of cis-HCOOH* was −0.19 eV, which was less negative than that of the trans one (−0.36 eV), implying that the N sites adjacent to the TM active sites played a vital role in improving the adsorption strength of HCOOH on the metal atom via the H-bonding formation.

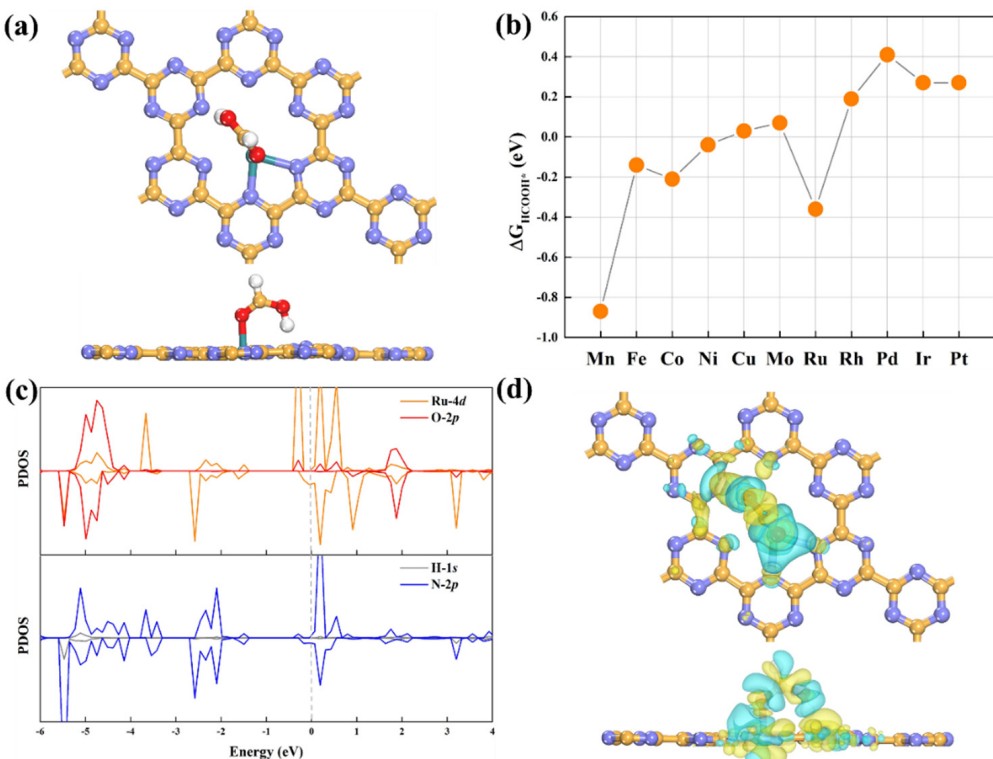

**Figure 2.** (**a**) The top and side views of HCOOH absorbed on Ru/g-CN structure after full relaxation. (**b**) The absorption free energies of HCOOH on TM/g-CN. (**c**) Projected density of states of the HCOOH molecule on Ru/g-CN. The Fermi level was set to zero, indicated by the dashed line. (**d**) The charge density difference of HCOOH absorbed on Ru/g-CN with an isosurface value of 0.003 eÅ$^{-3}$.

To better understand the HCOOH adsorption on TM/g-CN, the partial density of states (PDOSs) for HCOOH* on TM/g-CN was computed, where Ru/g-CN was again adopted as an example. As displayed in Figure 2c, an appreciable interaction could be observed between the Ru–4$d$ and O–2$p$ orbitals. Notably, limited hybridization between the H–1$s$ orbitals of the hydroxyl group and the N-2$p$ orbitals of g-CN could be identified, verifying the formation of the H–bonding between them. Additionally, significant charges accumulated at both N and Ru atoms to which HCOOH bound (Figure 2d), suggesting that HCOOH may have oxidized on Ru/g-CN. As a result, the O–H bond was stretched by approximately 0.03~0.10 Å due to the formation of hydrogen bonding between the hydroxyl-H and N atoms of the g-CN substrate.

After examining the adsorption of the HCOOH reactant, we explored the catalytic performance of these TM/g-CN candidates for the FAOR. According to a summary of previous studies [55–58], there are two distinct reaction pathways for the FAOR, i.e., direct and indirect pathways (Figure 3). In the direct pathway, HCOOH undergoes two successive dehydrogenation steps to reach the final $CO_2$ product: $HCOOH_{(aq)} \rightarrow HCOOH^* \rightarrow HCOO^*/COOH^* \rightarrow CO_{2(g)}$. As for the indirect pathway, HCOOH is firstly dehydrated to generate the CO* intermediate, followed by its oxidation to create the $CO_2$ product: $HCOOH_{(aq)} \rightarrow CO^* + H_2O \rightarrow CO_{2(g)}$. It was noteworthy that during the $CO_2$ formation, the cleavage of the O–H, C–H, and C–O bonds could be involved along the two reaction pathways. To examine the energetically favorable pathway for the FAOR in our proposed TM/g-CN catalysts, we took Ru/g-CN as a representative to compute the kinetic barrier for the O–H, C–H, and C–O bond splitting, which corresponded to the generation of the HCOO*, COOH*, and CO* intermediates, respectively. As expected, due to the sufficient activation of the O–H bond after HCOOH* on Ru/g-CN, the energy barrier for O–H cleavage to form HCOO* was only 0.04 eV, which was much lower than those of C–H (1.19 eV) and C–O (3.69 eV) cleavage (Figure S4), implying the preferential fracture of the O−H bond without the formation of poisonous CO. Thus, in the following discussion, we only focused on the direct pathway via the HCOO* intermediate for the FAOR on TM/g-CN candidates, which was consistent with previous studies [55,59].

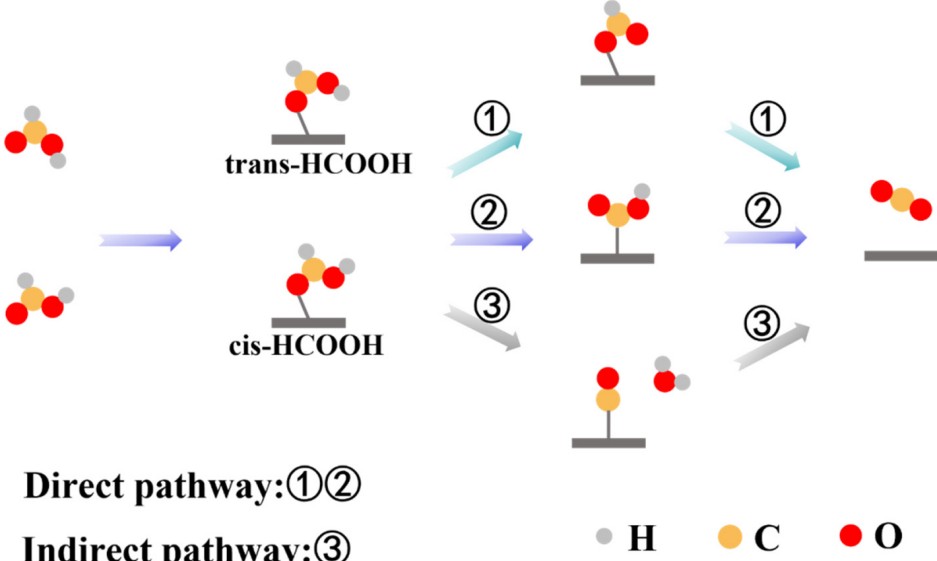

**Figure 3.** Schematic illustration of the direct pathway and indirect pathway mechanisms for FAOR on TM/g-CN.

Again taking Ru/g-CN as an example, we explored the FAOR catalytic activity. To this end, the computed free energy profile and the corresponding intermediates are presented in Figure 4. Starting from the HCOOH chemisorption with the ΔG value of −0.36 eV, it was dehydrogenated to form HCOO* via a facile O–H splitting process. Remarkably,

this step was slightly endothermic in the free energy profile by 0.15 eV. Meanwhile, in the formed HCOO* configuration, both of its O atoms bound with the Ru active site with lengths of 2.04 and 2.30 Å, respectively. Subsequently, the formed HCOO* could be further dehydrogenated to generate $CO_2$* with a ΔG value (0.08 eV). Notably, one C and one O atom of $CO_2$* adhered to the Ru site at distances of 2.27 and 2.03 Å, respectively, and the O–C–O bond angle was approximately 142°. Finally, the adsorbed $CO_2$* was released with a slightly negative ΔG value of 0.10 eV, suggesting that the formed $CO_2$ molecule could be effectively removed from the Ru/g-CN surface due to the weak interaction between them. Overall, from a purely thermodynamic perspective, among all elementary steps during the FAOR on the Ru/g-CN system, the dehydrogenation of HCOOH* to HCOO* could be viewed as the potential-determining step (PDS) due to its maximum ΔG of 0.15 eV, corresponding to the limiting potential of −0.15 V.

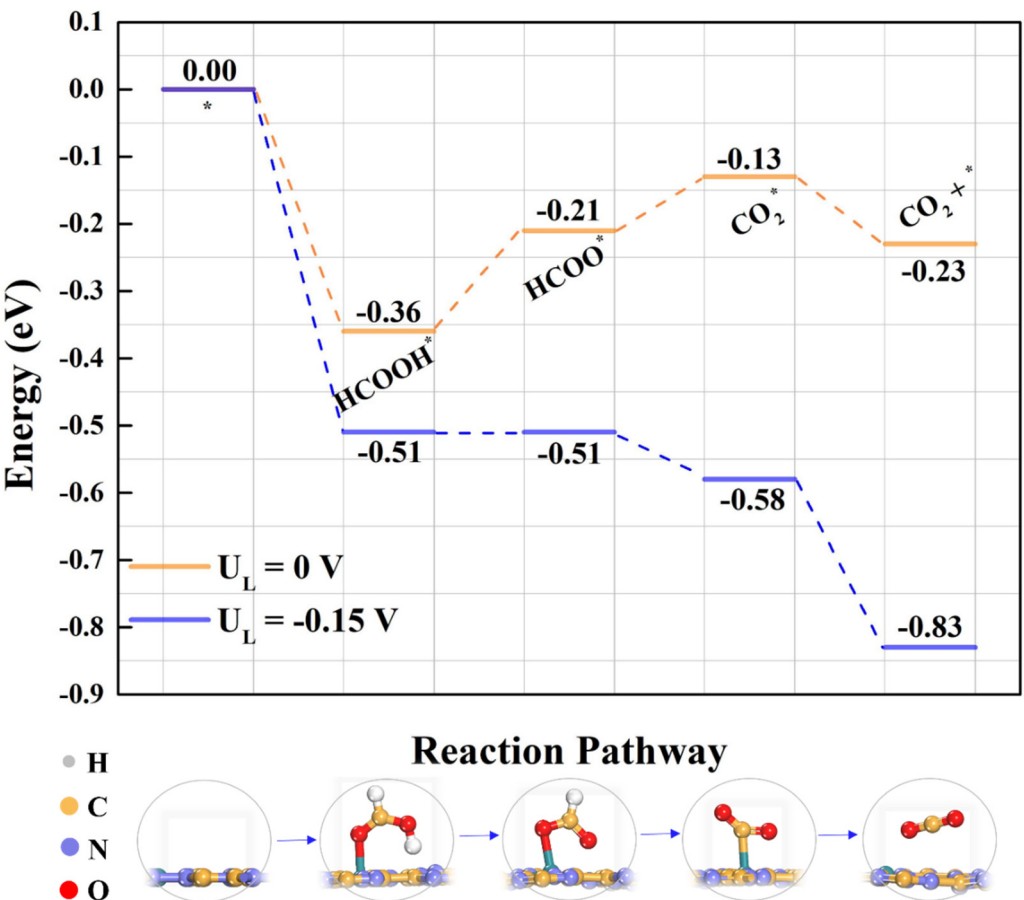

**Figure 4.** Free energy diagram of FAOR on Ru/g-CN via the direct pathway, and the corresponding structures of FAOR intermediates.

On the other hand, we computed the kinetic barrier for the FAOR on the Ru/g-CN system, which included three elementary steps (Figure S5): (1) HCOOH* → HCOO* + H*; (2) HCOO* → HCOO*′; (3) HCOO*′ → $CO_2$*. In detail, the first step was the O–H splitting, where the H* species was adsorbed on the N site of the catalyst with a rather low energy barrier of 0.04 eV. Then, the formed HCOO* rotated to give another HCOO*′ intermediate, in which the H atom of the C−H bond was pointing to the N atom with a distance of 2.49 Å. The energy barrier for this step was 0.71 eV, accompanied by an endothermicity of 0.22 eV. Next, the C−H bond was broken, in which the H atom was connected to the N site, and the remaining $CO_2$* species was adsorbed on the Ru site. For this process, the kinetic barrier was computed to be 0.89 eV. Thus, during the overall FAOR, the rate-determining step was the dehydrogenation of HCOO*′ due to its largest barrier of 0.89 eV, which was comparable

(or even lower) to those of some reported FAOR catalysts. Overall, Ru/g-CN exhibited high catalytic activity towards the FAOR considering both thermodynamics and kinetics.

In addition to the Ru/g-CN system, we also explored the FAOR catalytic activity on other TM/g-CN candidates by computing their free energy profiles. As shown in Figure S6, we found that the FAOR process on different TM/g-CN catalysts was hampered by different elementary steps. Similar to the Ru/g-CN catalyst, the HCOO* formation was the PDS on the anchored Co, Ni, Cu, Rh, Ir, and Pt catalysts, with the limiting potentials ranging from −0.29 to −0.69 V. In contrast, on the Fe/g-CN and Mo/g-CN surfaces, the dehydrogenation of HCOO* species hindered the whole FAOR process, and the computed limiting potentials were −0.22 and −0.41 V, respectively. As for Mn/g-CN, the interaction with HCOOH* ($\Delta G$ = −0.87 eV) was too strong, making the desorption of the final $CO_2$* product quite difficult due to the high energy input ($\Delta G$ = 0.91 eV), while Pd/g-CN displayed a weak capability to activate HCOOH* ($\Delta G$ = 0.41 eV), greatly limiting the subsequent dehydrogenation step. Based on the aforementioned results, we expected that Ru/g-CN would exhibit the highest FAOR catalytic activity among these TM/g-CN candidates due to its least negative limiting potential, which was even lower than those of other SACs [13,14].

### 2.3. Origin of Catalyst Activity

Understanding the catalytic trend of the FAOR on various TM/g-CN candidates could provide useful guidance for further designing and screening out efficient electrocatalysts. According to the famous Sabatier principle, the catalytic activity of a given catalyst is significantly determined by its interaction strength with reaction intermediates, in which an optimal binding strength leads to high catalytic activity [60,61]. To this end, we computed the adsorption energies of the involved FAOR intermediates, including HCOOH*, HCOO*, and $CO_2$*, on these TM/g-CN candidates. Interestingly, we found that there was an obvious linear scaling relationship between HCOOH* and HCOO*/$CO_2$* in these TM/g-CN systems, which could be written using the functions $\Delta G_{HCOOH*} = 0.53\Delta G_{HCOO*} - 0.14$ eV (Figure 5a) and $\Delta G_{HCOOH*} = -0.80\Delta G_{CO2*} - 0.24$ eV. Clearly, the FAOR catalytic performance was highly dependent on the adsorption energies of these reaction intermediate species. Since the PDS of the FAOR was highly dependent on the adsorption strength of HCOOH* and HCOO*, we plotted the variation of limiting potentials with a difference between the adsorption energy of HCOOH* and HCOO*, denoted as ($\Delta G_{HCOO*} - \Delta G_{HCOOH*}$). Amazingly, a volcano curve was obtained, in which Ru/g-CN was located at the peak of the volcano with the lowest limiting potential of −0.15 V and ($\Delta G_{HCOO*} - \Delta G_{HCOOH*}$) of approximately 0.18 eV (Figure 5b), implying its highest FAOR catalytic activity. In other words, the Ru/g-CN catalyst with a moderate binding strength with the two intermediates showed higher activity than the too strong (such as Mn/g-CN) or too weak (such as Pd/g-CN) catalysts.

To gain an insight into the remarkable differences in catalytic activity, we explored the intrinsic relationship between the adsorption energies of HCOOH* on TM/g-CN and the electronic properties of the TM active sites. To this end, the d–band center ($\varepsilon_d$) of the TM atom was then computed, which has been regarded as an ideal descriptor of the interaction trend between the reaction intermediates and the TM active sites. Interestingly, a good linear relationship between $\Delta E_{HCOOH*}$ and $\varepsilon_d$ was achieved (Figure 5c), in which the $\varepsilon_d$ value was more positive and the adsorption strength of HCOOH* was stronger. Similarly, there was an obvious scaling relationship between $\Delta E_{HCOOH*}$ and the polarized charges ($Q_{TM}$) on the TM active sites (Figure 5d). Thus, $\varepsilon_d$ and $Q_{TM}$ could be utilized as ideal descriptors to well rationalize the high FAOR catalytic activity of Ru/g-CN due to its optimal $\varepsilon_d$ and polarized charge.

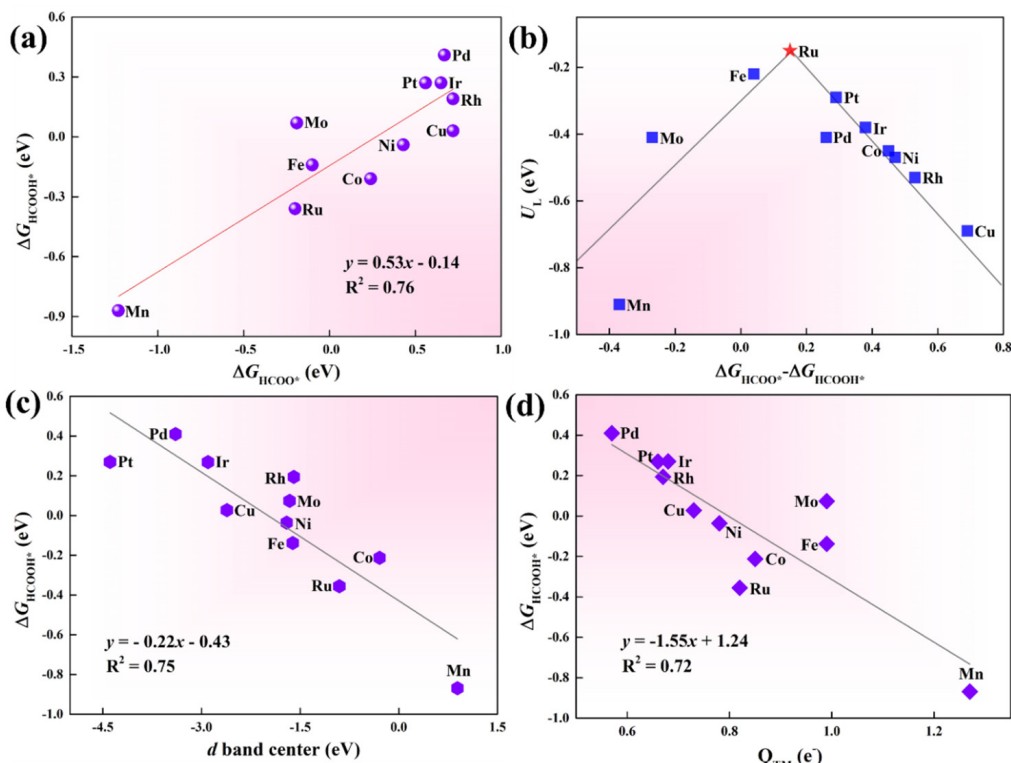

**Figure 5.** (**a**) $\Delta G_{HCOOH*}$ versus the $\Delta G_{HCOO*}$ species; (**b**) $U_L$ versus the free energy change of HCOO* and HCOOH* in each elementary step. The linear relationship between the adsorption free energy of HCOOH* and the corresponding (**c**) $d$ band center and (**d**) $Q_{TM}$.

## 3. Materials and Methods

All computations for the geometrical optimization and electronic properties were performed based on the spin-polarized density functional theory (DFT) method, as implemented in the DMol$^3$ code [62,63]. To describe the exchange–correlation interactions, the generalized gradient approximation (GGA) with the Perdew–Burke–Ernzerhof (PBE) function was employed [64]. The double numerical plus polarization (DNP) basis set was adopted, whose accuracy could be compared to that of Pople's 6–31G$^{**}$ basis set [65]. The DFT semicore pseudopotential (DSSP) method was used for the involved transition metals, in which their core electrons were replaced with a single effective potential by introducing some relativistic corrections [66]. The convergence criteria for the energy, maximum force, and displacement were set to $1.0 \times 10^{-5}$ Ha, 0.002 Ha, and 0.005 Å, respectively. The real-space global orbital cut-off radius was set as 5.2 Å for high accuracy. The possible van der Waals interaction between the FAOR intermediates and the catalysts was treated with the empirical correction in Grimme's method (DFT + D2) [67]. The Hirshfeld population analysis was employed to compute the charge transfer [68]. The linear synchronous transit/quadratic synchronous transit (LST/QST) tools in the DMol$^3$ code were used to locate the transition state [69]. The TM/g-CN catalysts were modeled by placing a single TM atom into the hole of a $2 \times 2 \times 1$ g-CN supercell. To avoid any interactions between the periodic images, a vacuum space of 20 Å was employed in the perpendicular direction. During the geometry optimization, a $5 \times 5 \times 1$ $k$-point was used, whereas the Brillouin zone was sampled with a denser $12 \times 12 \times 1$ centered k point grid for the electronic property computations. To evaluate the stability of these SACs on the g-CN substrate, we computed their binding energies ($E_{bind}$) using

$$E_{bind} = E_{TM/g\text{-}CN} - E_{TM} - E_{g\text{-}CN}$$

To explore the FAOR catalytic activity of these TM/g-CN catalysts, we computed the Gibbs free energy changes ($\Delta G$) of all possible elementary step with the computational hydrogen electrode (CHE) model [70–72], which has been widely used to describe the catalytic performance of a variety of electrochemical reactions, including the FAOR [13,14]. According to this method, the $\Delta G$ value could be determined with

$$\Delta G = \Delta E + \Delta ZPE + T\Delta S$$

where $\Delta E$, $\Delta ZPE$, and $\Delta S$ represent the differences in the DFT total energy, zero-point energy, and entropy between the reactant and the product, respectively, and the temperature T was set as 298.15 K. The values of the ZPE and S for freestanding molecules, such as HCOOH, CO, $CO_2$, $H_2O$, and $H_2$, were taken from the NIST database. As for the adsorbed reaction intermediates, however, their ZPE values could be obtained by performing harmonic vibrational frequency computations, for which their entropy contribution was neglected. Then, the limiting potential ($U_L$) was derived from the maximum free energy change among all elementary along the lowest-energy pathway

$$U_L = -\Delta G_{max}/e$$

According to this definition, a less negative $U_L$ value would require a lower energy input, thus, suggesting a higher FAOR catalytic activity. In addition, to simulate the realistic aqueous environments during the FAOR process, a conductor-like screening model (COSMO) was adopted with an $H_2O$ dielectric constant of 78.54 [73].

## 4. Conclusions

In summary, by performing DFT computations, we proposed TM-based SACs anchored on g-CN substrate as FAOR catalysts. Our results showed that these TM SACs could be firmly anchored on a g-CN monolayer, guaranteeing their high stability. Furthermore, by comparatively computing the kinetic barriers of C–H, O–H, and C–O cleavage, we found that the FAOR could proceed along a direct reaction mechanism on TM/g-CN catalysts. In particular, based on the computed free energy changes of all elementary steps, Ru/g-CN was revealed as a promising FAOR catalyst, with an ultralow limiting potential of $-0.15$ eV and a moderate kinetic barrier of 0.89 eV. Interestingly, the high FAOR catalytic activity of Ru/g-CN could be well rationalized through its moderate binding strength with the involved reaction intermediates due to its unique electronic properties, reflected by its moderate d-band center and polarized charge. Our findings not only offer cost-effective opportunities to develop SACs for a sustainable FAOR in the DFAFC, but also provide an in-depth understanding of the catalytic mechanism and origin of the FAOR, which could be conducive to further develop stable, low-cost, and highly efficient electrocatalysts for the FAOR.

**Supplementary Materials:** The following supporting information can be downloaded at: https://www.mdpi.com/article/10.3390/catal13010187/s1, Figure S1. The top and side view of the optimized g-CN and the corresponding coordination information. Figure S2: variations of temperature and energy as a function of time for AIMD simulations of Ru/g-CN; inset is top view of the snapshot of atomic configuration. The simulation was run under 500 K for 10 ps with a time step of 1 fs. Gray, blue, and dark green spheres represent the C, N, and Ru atoms, respectively; Figure S3: the band structure of all TM/g-CN that we considered; Figure S4: energy barrier profiles for HCOOH* − COOH* and HCOOH* − $CO_2$* + $H_2O$ on Ru/g-CN; Figure S5: energy barrier profiles for HCOOH decomposition on Ru/g-CN; Figure S6: the free energy diagram of FAOR on TM/g-CN along direct pathway.

**Author Contributions:** Y.L. and J.Z. outlined the work plan; M.Y. and A.Q. conducted the computations; M.Y. drew the figures and drafted the manuscript. Q.C. revised the drafted manuscript. All authors participated in the reviewing and publication processes of the article. All authors have read and agreed to the published version of the manuscript.

**Funding:** This work was financially supported by the Natural Science Funds (NSFs) for Distinguished Young Scholars of Heilongjiang province (No. JC2018004).

**Acknowledgments:** The authors acknowledge the support of this study by the Natural Science Funds (NSFs) for Distinguished Young Scholars of Heilongjiang province (No. JC2018004).

**Conflicts of Interest:** The authors declare no conflict of interest.

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
