# Peer review of "Computational Study on the Catalytic Performance of Single-Atom Catalysts Anchored on g-CN for Electrochemical Oxidation of Formic Acid"

_catalysts, doi:10.3390/catal13010187_

Round 1

Reviewer 1 Report

In this work the authors presented a computational study on metal single atom catalysts embedded in carbon nitride for the oxidation of formic acid to CO2  from first principle simulations.

 The work is in my opinion of interest for the catalysis audience, but the current version of the work need to be revised according to following two points:

1) C3N4 nanosheets assume a corrugated structure rather than being flat [10.1021/acs.chemmater.6b02969; 10.1039/D1CY00451D ; 10.1021/acscatal.1c05610; 10.1016/j.jcat.2022.12.014]. This fact should be mentioned in the main text. Importantly, the authors should motivate their choice of modelling the flat model (Figure 1a).

2) It is unclear to me if all reaction pathways have been explored for all catalysts, or if the picture was restricted somehow. In this respect, it is unclear also if scaling relationship plots (Figure 5) arise from metal species undergoing the same reaction path, or if the results are obtained by different paths.

Author Response

Comment 1: C3N4 nanosheets assume a corrugated structure rather than being flat [10.1021/acs.chemmater.6b02969; 10.1039/D1CY00451D; 10.1021/acscatal.1c05610; 10.1016/j.jcat.2022.12.014]. This fact should be mentioned in the main text. Importantly, the authors should motivate their choice of modelling the flat model (Figure 1a).

Reply: We are very grateful for the referee’s invaluable suggestion. Indeed, there are two possible structures for C3N4 nanosheet, including corrugated and flat ones, in which the former is more stable as mentioned by the referee. However, it is worthy to pointing that the adopted substrate for SACs in this work is g-CN, not g-C3N4. According to previous reports [42-47], the g-CN monolayer has a flat structure and is not prone to deformation. Thus, in this work, we adopted the flat structure for g-CN.

Comment 2: It is unclear to me if all reaction pathways have been explored for all catalysts, or if the picture was restricted somehow. In this respect, it is unclear also if scaling relationship plots (Figure 5) arise from metal species undergoing the same reaction path, or if the results are obtained by different paths.

Reply: We are very grateful for the referee’s invaluable suggestion. According to the suggestion, in the revised manuscript, we have addressed this question and please see line 133, “After examining the adsorption of HCOOH reactant, we then explored the catalytic performance of these TM/g-CN candidates for FAOR. According to a summary of previous studies [48-51], there are two distinct reaction pathways for FAOR, i.e., direct and indirect pathways (Fig. 3).”, in which we considered all reaction pathways for all catalysts; Also please see line 145, “Thus, in the following discussion, we only focused on the direct pathway via HCOO* intermediate for FAOR on TM/g-CN candidates, which is well consistent with previous studies  [48,52].”, indicating that the figure 5 are obtained by the same reaction path of direct via HCOOH(aq) HCOOH* HCOO* CO2(g).

Reviewer 2 Report

In this manuscript, Qadeer et al. provided a theoretical study based on DFT calculation on various types of single-atom catalysts (transition metal single atoms coordinated with graphitic carbon nitride) for catalysing the electrochemical oxidation of formic acid. Overall, this work has good novelty and the manuscript is well organized. Based on the expertise of this reviewer, I would in general support the publication at the journal Catalysts. However, some technical issues need to be resolved before the possible acceptance of the manuscript. Please see below for more detail.

1. For consistency purposes, figures need to be improved. Figure 1a and c, the same atoms were represented by balls with different colours. The same applies for Figure 2a and d.

2. Some technical terms should be defined, for example, UL in Figure 5b.

3. In Introduction, recent works on single-atom catalysts are suggested to be referenced (e.g., Materials Reports: Energy, 2022, 2, 100144).

4. Figure 4, for the schematic illustration of the reaction pathway, the various elements should be labelled.

5. The English writing needs to be improved. For instance, (1) line 58, “we noted that there few studies”, double check the grammar for this sentence; (2) line 330, “in-deep” should be revised into “in-depth”.

Author Response

Comment 1: For consistency purposes, figures need to be improved. Figure 1a and c, the same atoms were represented by balls with different colours. The same applies for Figure 2a and d.

Reply: We are very grateful for the referee’s invaluable suggestion. In the revised manuscript, we have updated the Figure 1c and 2d to apply the same colors with 1a and 2a, please see line 128 and 158.

Comment 2: Some technical terms should be defined, for example, UL in Figure 5b.

Reply: We are very grateful for the referee’s invaluable suggestion. According to the suggestion, we have defined the UL in Materials and methods section line 245, “the limiting potential (UL) was derived from the maximum free energy change among all elementary along the lowest-energy pathway.” Accordingly, we have changed all the onset potential to limiting potential in the revised manuscript.

Comment 3: In Introduction, recent works on single-atom catalysts are suggested to be referenced (e.g., Materials Reports: Energy, 2022, 2, 100144).

Reply: We are very grateful for the referee’s invaluable suggestion. According to the suggestion, in the revised manuscript, the format of Ref. 28 has been updated and please see line 335, “[28] Abdelghafar F, Xu X, Shao Z. Designing single-atom catalysts toward improved alkaline hydrogen evolution reaction. Mater. Rep.: Energy 2022, 2, 100144.

.

Comment 4: Figure 4, for the schematic illustration of the reaction pathway, the various elements should be labelled.

Reply: We are very grateful for the referee’s invaluable suggestion. In the revised manuscript, we have re-plotted the Figure 4 to label the different elements with different colors, please see line 175.

Comment 5: The English writing needs to be improved. For instance, (1) line 58, “we noted that there few studies”, double check the grammar for this sentence; (2) line 330, “in-deep” should be revised into “in-depth”.

Reply: We are very sorry for these errors. In the revised manuscript, we have corrected these mentioned “we noted that there few studies” to “we noted that there are few studies” and “in-deep” to “in-depth”, respectively. Please see line 47 and 260.

Round 2

Reviewer 1 Report

I appreciate the effort profused by the authors in improving their work.

However, I feel that point 1 of my previous report remains opened.

The authors replied that: "Indeed, there are two possible structures for C3N4 nanosheet, including corrugated and flat ones, in which the former is more stable as mentioned by the referee. However, it is worthy to pointing that the adopted substrate for SACs in this work is g-CN, not g-C3N4. According to previous reports [42-47], the g-CN monolayer has a flat structure and is not prone to deformation. Thus, in this work, we adopted the flat structure for g-CN."

In the abstract the authors report that: "In this work, taking single atom catalysts (SACs) supported on graphitic carbon nitrides (g-CN) as potentials catalysts".

In addition, figure 1 reports clearly a carbon nitride nanosheet. Therefore, why the authors say in the reply that their substrate is not C3N4 but g-CN? Also, what exactly is g-CN? Is it carbon nitride, N-doped graphene, etc?

In addition, previous literature showing that carbon nitride prefer to stay corrugate should be mentioned, and hence the motivation behind the authors' choice should be given clearly.

Author Response

Comment 1: The authors replied that: "Indeed, there are two possible structures for C3N4 nanosheet, including corrugated and flat ones, in which the former is more stable as mentioned by the referee. However, it is worthy to pointing that the adopted substrate for SACs in this work is g-CN, not g-C3N4. According to previous reports [42-47], the g-CN monolayer has a flat structure and is not prone to deformation. Thus, in this work, we adopted the flat structure for g-CN."

In the abstract the authors report that: "In this work, taking single atom catalysts (SACs) supported on graphitic carbon nitrides (g-CN) as potentials catalysts".

In addition, figure 1 reports clearly a carbon nitride nanosheet. Therefore, why the authors say in the reply that their substrate is not C3N4 but g-CN? Also, what exactly is g-CN? Is it carbon nitride, N-doped graphene, etc?

In addition, previous literature showing that carbon nitride prefer to stay corrugate should be mentioned, and hence the motivation behind the authors' choice should be given clearly.

Reply: We are very grateful for the referee’s invaluable suggestion. In the revised manuscript, we have addressed this question. Please see page 2, line 32, “Recently, g-CN has been successfully fabricated via the reaction of cyanuric chloride and sodium by a simple solvothermal method [42]. Notably, different from the corrugated configuration of the well-known g-C3N4 [43-46], all atoms within g-CN framework are in exactly the same plane (Fig. S1), which is well consistent with previous reports [47-52]. In other words, the configurations of CxNy nanosheet is highly dependent on the ratio between C and N [53]. Simar planar configuration can be also observed for C2N monolayer [54].” In addition, we have provided the information of CONTCAR file on g-CN, from which it can be clearly observed that the ratio between C and N is 1:1, namely, g-CN monolayer, rather than other CxNy monolayer. Also, we have added some related references, including Ref. 42-46, and 54.

Round 3

Reviewer 1 Report

The authors addressed the mentioned point. The work can be published.